# Natural Environments in University Campuses and Students’ Well-Being

**DOI:** 10.3390/ijerph21040413

**Published:** 2024-03-28

**Authors:** Helena Ribeiro, Keila Valente de Souza Santana, Sofia Lizarralde Oliver

**Affiliations:** Department of Environmental Health, Faculdade de Saúde Pública, Universidade de São Paulo, São Paulo 01246-904, Brazil; keilla@usp.br (K.V.d.S.S.); s.oliver@usp.br (S.L.O.)

**Keywords:** mental health, well-being, university campus, greenness, nature-based solutions

## Abstract

Most recent university campuses follow the North American model, built on city limits or countryside, with large separate buildings in open green spaces. Studies suggest that the prevalence and severity of mental health issues among university students has been increasing over the past decade in most countries. University services were created to face this growing problem, however individual-based interventions have limited effects on mental health and well-being of a large population. Our aim was to verify if and how the natural environment in campuses is focused on programs to cope with the issue of mental health and well-being among students. A systematic review of literature was undertaken with search in Scopus and LILACS with the keywords “green areas” AND “well-being” AND “Campus”, following PRISMA guidelines. As a result, 32 articles were selected. Research on the topic is recent, mostly in the USA, Bulgaria, and China. Most studies used objective information on campuses’ greenness and/or university students’ perception. Mental health was usually measured by validated scores. Findings of all the studies indicated positive association between campus greenery and well-being of students. We conclude that there is a large potential for use of university campuses in programs and as sites for students’ restoration and stress relief.

## 1. Introduction

Most of the more recent university campuses follow the North American model. They are built on the limits of the city or the countryside, with large separate buildings, implanted in open green spaces, along avenues and streets [1].

Since the beginning of the twentieth century, in the field of pedagogy, the New School movement has spread throughout the world. This movement demanded a profound renewal in the school environment, associated with a change in the way the teachers exercised authority. The proposed changes included transforming school buildings into more horizontal ones, so that they would become more open to nature [2]. In kindergarten and primary education, the principles of free movement of children and access to the outdoors and nature have been adopted in many countries. In Brazil, Anísio Teixeira defended the park-school with two periods, in which one of the periods, in a large, wooded space, children performed physical, social, and artistic activities [3]. Teixeira was a mentor to educators, such as Darci Ribeiro, who became Minister of Education.

At the university level, these principles were not so evident, but the heritage of the architectural design, with horizontal buildings and large green areas between them, was maintained.

University campuses, the majority characterized by having extensive lawns and green spaces, cause large expenditures for their maintenance, but are scarcely used as recreation or leisure areas for the well-being and mental health of students or the surrounding community. There is a large potential for their use as a natural environment for the well-being of students, teaching, office staff, and the general community.

On a university campus, greenness can include trees, vegetation, lawns, bushes, etc. that flank pathways and roads, and exist in spaces between buildings [4], as well as water bodies comprising blue spaces [5].

Student quality of life is a complex composite of physical health, psychological state, and relationships with the surrounding cultural, social, and environmental context [6], and it is an important precursor for learning and success in college [4].

University students face a series of academic, interpersonal, financial, and cultural challenges that may cause psychological and other health problems. Students’ stress may result also from difficult daily targets, long study hours, bad relationships with classmates, emotional distress, loss of self-identity, and low self-esteem [7]. Many of the young adults who attend universities face their first prolonged period away from home [8]. This change in environment, combined with academic, social, and financial pressures, can be stressful, with impacts on the physical status and wellness of students [8,9]. Ibes et al. reaffirm that the mental health crisis is particularly pronounced on college campuses, where students encounter life stressors, such as academic and extracurricular demands, relationships, financial concerns, family expectations, identity development, and racial and cultural differences [9]. Digital overload exacerbates chronic stress and anxiety, as college students tend to spend many hours a day on smartphones, tablets, computers, and television [9]. Among young adults, intense technology use may exacerbate symptoms of stress, fatigue, sleep disorders, and depression [9]. Students that do not live on campus also tend to spend the majority of their time on college campuses, more waking hours than they spend at home, due to academic tasks [4,7].

Studies suggest that the prevalence and severity of mental health issues among university students has been increasing over the past decade in North America [10]. According to the Spring 2022 data report from the National College Health Assessment of the US, 75% of college students experienced moderate to serious psychological distress, and 48% felt lonely [11]. In China [7], India [12], Canada [13], and Korea in 2019, 45% of the students suffered from depression [14], and in many other countries around the world [4], it is becoming a widespread problem that was aggravated during the COVID-19 pandemic [15,16]. This trend is apparently similar in low- and middle-income countries (LMIC). For example, in Brazil, young adults (18 to 29 years old) represent the group with the lowest percentage of depression (5.9% in 2019) but presented the largest increase in diagnoses of depression from 2013 to 2019, 51%. Suicide rates among the age group 20 to 39 increased from 6.73 to 8.19 per 100,000 inhabitants in the same period [17].

According to the WHO, the most common mental health disorders, because of their high prevalence, are depression and anxiety. The first is characterized by symptoms such as sadness, low interest or pleasure, feelings of guilt or low self-esteem, changes in sleep or appetite pattern, sensation of tiredness, and low concentration. The second presents symptoms as anxiety, fear, panic, phobias, obsessive-compulsive behavior, and post-traumatic stress [18]. On top of these two common disorders, suicidal ideation and suicide attempts have become a serious problem among university students. The transition between childhood and adulthood is characterized by rapid physical, psychological, and cognitive developments and lays the foundation for future mental health and resilience. An estimated 70% of mental health problems first appear during this development period [19].

Many university services have been created to face this growing problem; however, interventions based in individuals have limited effects on the mental health and well-being of a large population. Besides, although many universities offer mental health treatment and counseling programs, a vast majority of students are reluctant to use them due to personal and public stigma [20]. Ibes et al. [9] reinforce that universities may address this crisis in traditional ways, such as counseling staff and mental health programming. Yet, an accessible, simple, and affordable tool for relieving stress, such as greenspace, is overlooked. Considering that mental health issues are complex and need a holistic approach, Windhorst and Williams [10] argue that strategies should include campus environments, such as social inclusivity, students’ safety, support services, but also the natural environment as an important dimension of mental health. In their view, this absence is striking in studies and in real health policies in Latin America [10].

Our first aim was to focus our study on Latin America’s campuses, as we teach and study in a Latin American context where young adults from low- and middle-income families are increasingly going to university, and where mental health care is underfunded. Therefore, alternative preventive strategies should be explored. However, our first literature search in Scopus and Web of Science showed no studies on the relationship between university students’ well-being or mental health and green spaces.

Wolf et al. [20], in their scoping review of literature on Urban Trees and Human Health, pointed out that of 201 studies analyzed, only one was based in Mexico, and 1% were undertaken in South America [20]. This indicates that studies on the effects of the natural environment on health are still scarce in Latin America.

According to Liu et al. [7], there are two main theoretical perspectives related to the restorative effects of natural environments: (a) The psycho-evolutionary theory holds that viewing the natural environment can improve the mood of people and mitigate stress by blocking out pessimistic thoughts and ruminations. (b) The Attention Restoration Theory (ART) holds that intensive or prolonged focus leads to exhaustion of mental capacity and that directed attention capabilities can be restored by contact with nature [7]. According to ART, the natural environment should exhibit four types of characteristics to assist in this process: a sense of escape or being away from daily’s routine, extent to invoke imagination [as in a forest trail], fascination, and be compatible with the desires of the individual. Maximum benefit is obtained when all four are present [21]. Malekinezhad et al. [22] highlight that based on the Stress Restoration Theory (SRT) and ART, restorative outcomes of natural environments have been operationalized into dimensions of “direct attention restoration”, “clearing random thoughts”, and “relaxation and calmness”. Bardhan et al. [11] consider that university students may benefit particularly strongly from nature exposure because of their high academic and social demands in preparing for professional careers. Some authors emphasize that, according to the biophilia hypothesis, humans have developed and maintained an affinity for nature throughout their evolution, which makes positive outcomes from exposure to green spaces on the human body and mind stand to reason [11,23]. Other authors emphasize mediators of this process as reduced exposure to air pollution, heat island, and noise, as well as encouraging health-enhancing behaviors such as physical activity and social interactions [5]. Other authors reinforce the salutogenesis concept, focusing on factors to promote and support human health and well-being and recognizing contact with nature as a stress-reducing health promotion tool [24].

Having a hypothesis that campus green spaces represent a valuable resource for promoting mental health among university students, we aimed to verify if and how the natural environment is approached in scientific literature on well-being in university campuses, and how it is being focused on programs and policies to cope with the growing issue of mental health among university students. Even though, in recent years, this topic has been the focus of many dispersed studies, a broad review of literature on the theme is non-existent, thus it is innovative and can point to the main findings and gaps. To address this objective, a systematic literature review was undertaken to assess the extent, range, and nature of research in this topic area.

## 2. Materials and Methods

The method adopted was a systematic review of literature following Prisma guidelines. The search was undertaken with no date range, in the basis Web of Science, Scopus, LILACS, and SciELO, because the focus of the last two is in Latin America, with the keywords: “green areas” AND “well-being” AND “Campus”, and variations with the words “green areas” for “greenness”, as well as translation to the words to Portuguese: “áreas verdes” AND “bem-estar” AND “campus universitário”, and Spanish: “espacios verdes” AND “bien-vivir” AND “campus universitário”. First results indicated 7 documents in Scopus, none regarding university students or young adults, and 49 in Web of Science, of which 16 focused on the relationship of mental health and greenspace in a Latin American context, but none focused on college students or young adults.

As no manuscript was found regarding studies in Latin America, our investigation focused on any region of the world. A Boolean search was performed on Scopus, LILACS, and SciELO to find publications with no initial cut-off date until February 2023. We did not use an initial cut-off date because we wanted to verify when studies on the theme started to appear in scientific literature. Data were collected from Scopus, which includes Medline and Embase databases, and demonstrated adequate results. LILACS and SciELO brought poor results. The descriptors and Boolean operators were three search strings, which were later systematized: “greenness” OR “green areas” AND “university campus” called Search 1; the second: “greenness” OR “green areas” AND “mental health” called Search 2; and the third: “university campus” AND “mental health”, called Search 3.

The inclusion criteria used were the simultaneous presence of the terms of each search in the title, or in the abstract, or in the keywords. As exclusion criteria, the following were stipulated: absence of simultaneity of the terms of each search in the title or abstract or in the keywords, articles with a study population outside the young adult age group, articles that dealt with botanical or faunal issues, sanitation, or other aspects of the university campus without association with students’ mental health or well-being. Articles that associated greenness with health issues unrelated to well-being or mental health were also discarded.

The three searches were necessary because the use of the three words concomitantly did not return any results, even without the limitation of a geographical area. The searches were separated and the relationships between their results were made manually. The analysis technique was reading and documentary analysis by the three authors, the instrument used was an Excel sheet organized in a table, and the relevant information extracted manually was authors, region of study, year of publication, objective of study, method used, sample, and results obtained.

After reading the abstracts obtained in the three searches by the three authors, 36 articles were selected by consensus, of which 4 were discarded, 2 of them because they were book chapters, 1 because it was an abstract in meeting proceedings, and 1 article whose text was not available, leaving 32 studies for analysis.

The PRISMA guidelines [25] were used to organize and systematize the search and results, as shown in Figure 1.

## 3. Results

As described in the methods, in total, 32 articles were found. Of these, only two studies were published before the year 2015, in China (2009) and the United States (2014), showing that it is a recent topic in the scientific literature.

In 2016 and 2017, one paper was published in the U.S. and one in South Africa. From 2018, six studies were found, four in Bulgaria, one in the USA, and one in China. From 2019, three studies in Bulgaria, Turkey, and Canada. From 2020, there were five studies, with expansion to other countries. The theme, of recent interest, is also limited to a few countries in the world, as can be seen in Figure 2 and Figure 3. The United States, Bulgaria, and China stand out in terms of number of studies, with the exception that in Bulgaria, the studies are concentrated in one research group.

Results were also analyzed following PRISMA guidelines and are presented in Table 1 indicating the following aspects: year of study, authors and reference number, study location, objective, method, sample, and results.

## 4. Discussion

Research on the topic is recent, mostly in the northern hemisphere, and in higher-income countries. Nine studies (28%) were undertaken in low- and middle-income countries (LMIC), where university degree expansion is taking place. No studies were undertaken in Latin America, showing that the focus on campus environment for the growing issue of mental health among university students is unexplored. The pressure on college students is as hard as, or even harder, than in higher income countries, due to economic hardships, and the need to work from an early age, in most cases concomitantly with academic life. Another finding is that studies are often concentrated in a research group. Among the 32 articles, nine were from repeated authors.

The COVID-19 pandemic highlighted the importance of open space for university students and seemed to motivate studies on the topic in more countries and areas, as most manuscripts date from 2020 on.

Regarding methods for identifying and calculating the surrounding natural environment, most studies used objective information on campuses’ greenness, based on satellite images with Normalized Difference Vegetation Index-NDVI (ten studies), vegetation index, tree cover density, and land cover database. Some research employed objective and subjective measures such as university students’ perception on the presence and quality of greenspace in campus, and a few around their residential area. Some used only perceptive values on greenness. One study used a NatureDose app with GPS and a phone sensor in the student cell phone [11], and one study was a pre-post-test randomized experiment with a 360-degree video platform via an online survey [16]. The research by Reese et al. [38] used a photovoice technique with a photvoicekit to foster students to take images on class outings, then later host a photo exhibit on campus and use their vision for discussing a variety of ways that underdeveloped greenspaces on campus might be utilized to promote student physical and mental health [38]. Three of the articles described results of interventions on campus such as the creation of a sensory garden [40], therapeutic sensory garden [38], short greenspace interventions [9,37], and forest activities [14].

Mental health was mostly measured by surveys or questionnaires (21 studies, 65%) with validated scores. Ning et al. [35] used physiological indicators to measure restorative effects of exposure to natural campus areas (green and blue spaces): systolic blood pressure, diastolic blood pressure, and heart rate recovery amounts [35]. Asim et al. [12] measured brainwaves with EEG to assess the benefits, and Souter-Brow et al. [24] measured salivary cortisol of students.

The main findings of all but one [19] of the studies pointed to a positive association between campus greenery and well-being of students, indicating the restorative role of greenness and water bodies on campus and/or around students’ residential areas. One of the 32 studies, undertaken in a natural desert environment, found a protective role of brownness (35% decrease in the incidence of depression), which may be explained by scarce green areas and familiarity with the desert environment and color [29]. In 12 of the studies, authors focused on green spaces’ settings and type of nature in association to mental well-being. High natural attributes [7], high biodiversity [19], botanical garden [22], vegetation novelty/uncertainty [2], expansive vistas [2], natural sound and acoustic ambiance [2,19], and blue space [32] indicated higher levels of restoration and lower negative moods. Closeness to buildings [4], window views overlooking greenery and sky, and also plants indoors [12] were associated with lower risk of depression and of severe anxiety. Quality of green space showed greater importance to mental health than use and quantity [35], while grayness of buildings was associated with an increase in depression [28].

Nine of the articles described results of interventions with students aiming to verify if natural environments could improve their mental health and well-being. Three of them used living experience in healing [40] and sensory gardens for quality of life [38], stress reduction, well-being and productivity [24]. Forest activities [14], trail and digital detox [9], and nature exposure were associated with positive moods [11] and to decreased test anxiety [26]. Brief nature exposure [35] indicated better heart indicators in water-front space, dense forest, and sparse forest.

Some mediating effect variables that explain how a greenspace affects beneficial outcomes were observed in a few studies: reduced road traffic noise, heat island, perceived air pollution, and isolation, as well as increased physical activity and social cohesion [5,31,33]. One of the studies highlighted those preventive strategies, dedicated mental health services, stable employment, and green infrastructure [27].

Most studies had cross-sectional design or were investigations on limited experiences. Evaluation of large-scale interventions to improve mental health, based on use and or improvements on green/blue spaces, with longitudinal methods, are still very limited to healing gardens and short periods of time.

Nevertheless, many students participated in these investigations. Adding students from all the samples came to a total of 27,955 individuals of both sexes and different academic backgrounds, in diverse areas around the world, from public and private universities or colleges. These high numbers make the findings quite robust.

We conclude that there is a large potential for use of university campuses in programs and as sites for students’ restoration and stress relief. Examples are therapeutic sensory gardens [24,37], forest activities [14], indoor plants and window views for plants and trees [12], green micro-breaks [9], walking and bicycling trails along the campus green areas, and benches in silent and shady gardens. Botanical gardens were also reported as missing to attract students’ experience for a restorative environment. Gulwadi [4] concluded that the space immediately next to buildings in which instruction takes place gains more significance when students spend longer hours in class, as that offers the closest access to greenspaces and enables social support [4]. Cultural aspects and geographical locations are of great importance according to the biophilia theory. Students’ perception of the wellness provided by greenness depends also on their experience, background, and taste. An important requirement is the sense of security of being outdoors and for this, care and maintenance of the campus green spaces are crucial. The costs of these items can be high, but are lower than those of expanding mental health services [41].

We highlight the relevance of students’ participation and engagement in activities for promoting student wellness and mental health, but also on planning land and nature use [38], on promoting campus sustainability [40], and natural area stewardship. Students’ perspective is particularly important in the face of this serious mental health crisis [41], as it may foster student trust, social cohesion [37], engagement, and learning that can be leveraged beyond a particular crisis to support longer-term sustainability goals.

University campuses might provide a salutogenic influence, as natural environments provide restoration to university students and are also a contributing factor to student retention because those who suffer mental health disorders are more likely to drop out, underperform academically, and are less likely to secure higher level employment [36].

Causes of psychological stress among university students are (a) academic: high academic demands, difficult daily targets, long study hours, different teaching methods with more responsibility to students, test anxiety, and extracurricular activities; (b) interpersonal: new relationships and friends, identity development, bad relationships with classmates, emotional distress, loss of self-identity, low self-esteem, prolonged period away from home, thus lower family support; (c) financial: pressures for paying tuition, living expenses such as food and academic material such as books, financial and professional concerns for the future, and family expectations; (d) cultural: racial and cultural differences, digital overload, long hours on campus for academic tasks and/or living on campus.

Nature contact can be a stress-reducing health promotion tool and a protective factor for mental health, based on the Stress Restoration Theory (SRT), by clearing random thoughts and lowering rumination; relaxation and calmness, and by restoring attention according to Attention Restoration Theory (ART). Pathways found in the studies include students’ perception; systolic and diastolic blood pressure, and heart rate; normalized average alpha brain waves measured with EEG; sensation of relief from stress; salivary cortisol; reduced scores on the anxiety and depression scales; increased mood, stress response and subjective well-being; and enhanced social life and social cohesion.

Thus, campus natural environments might restore psychological capabilities, improve attention, improve mood, and reduce stress and anxiety, with clinical evidence. In addition, programs and interventions on campus natural environments might build capabilities such as active life, adequate weight, social life, improved cognitive abilities, and personal growth.

Cox et al. [42] explain the four progressive levels of restoration: cleaning thoughts, recharging the capacity of directing attention, reducing internal noise, reflecting on life, its priorities, possibilities, actions, and objectives. Optimal levels of nature are not a silver bullet for the prevention and treatment of all mental health problems, nor can they act on the main causes of students’ stress, but they are an approach that can and should be applied for young adults in initial stages of depression and anxiety, and in conjunction with medical and social services, and increasing community-oriented actions.

Windhorst and William [10] claim for three actions: first to raise awareness of nature benefits among university students and staff; second, to make interventions on the campus to improve its attractiveness for community use and for students’ well-being; third, to develop nature-based therapies. This review article showed their importance and provided tools for these actions.

Gaps in the literature that need academic attention to deepen our understanding of the impact of green spaces on students’ well-being include: longitudinal studies, long-term interventions, psychology outcomes, and depression symptoms such as sadness, low interest or pleasure, feelings of guilt or low self-esteem, changes in sleep or appetite pattern, sensation of tiredness, and low concentration; anxiety symptoms as anxiety, fear, panic, phobias, and obsessive-compulsive behavior; suicide ideation and suicide attempts among students and how they could benefit from nature and nature activities.

Greenness in university campuses is even more important in LMIC countries, as a large proportion of the urban young adults live in neighborhoods crowded with small houses where vegetation is almost non-existent. We urge Latin American universities to turn eyes to this topic of investigation and practice.

We also call for the need of an interdisciplinary approach to the mental health issue among university students, with the integration of mental health services, landscape architecture, geography, campus administration, cultural activities, and physical activity practices.

## 5. Conclusions

Findings of all the studies indicated positive associations between campus greenery and the well-being of students. However, university campuses’ natural environments are largely under-utilized in programs and sites for students’ restoration and stress relief. As they usually have large green areas between buildings, some low-cost interventions and open-air programs could contribute to diminish the prevalence and severity of mental health disorders, such as depression and anxiety, among university students, together with other initiatives. Investigations among university students are still limited but showed a large potential also for student engagement and building of social capital.

## Figures and Tables

**Figure 1 ijerph-21-00413-f001:**
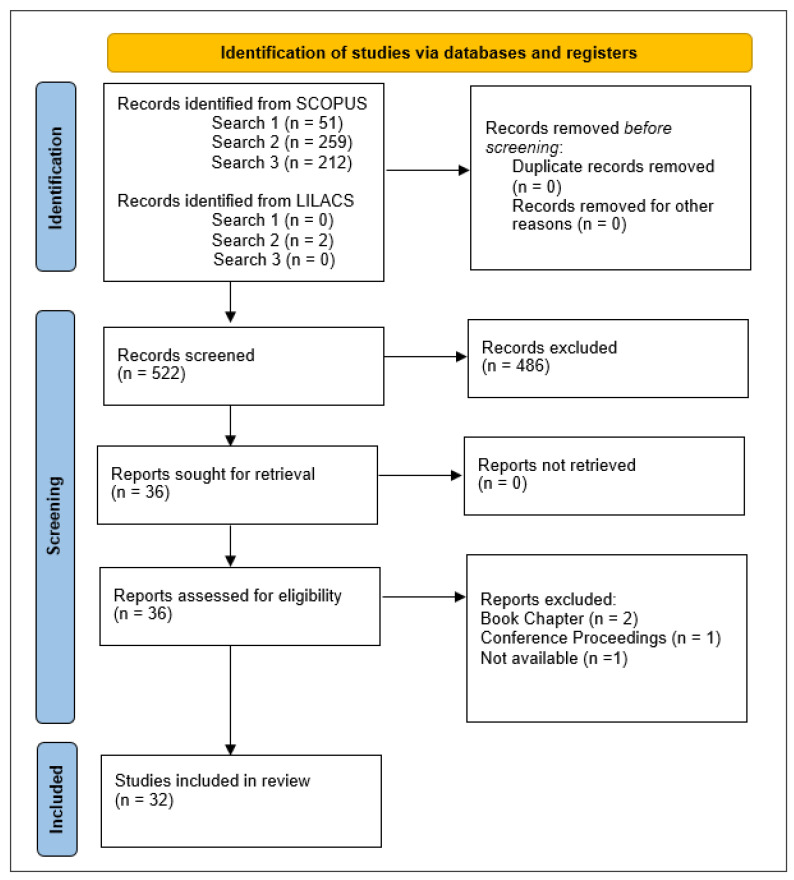
Search and results according to PRISMA guidelines.

**Figure 2 ijerph-21-00413-f002:**
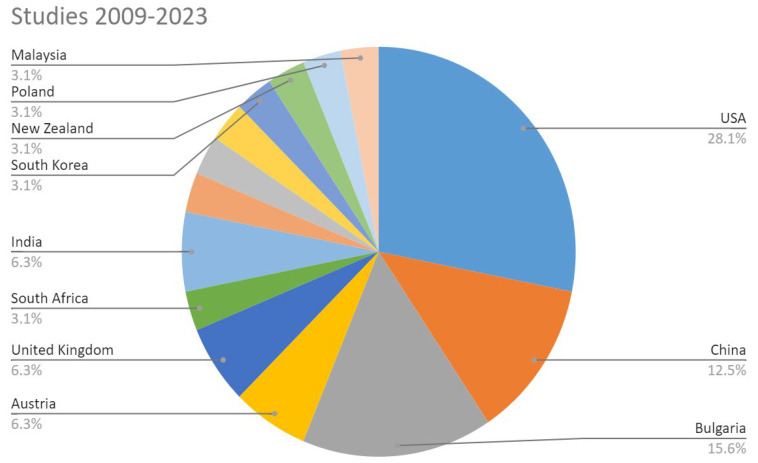
Studies by country 2009–2023.

**Figure 3 ijerph-21-00413-f003:**
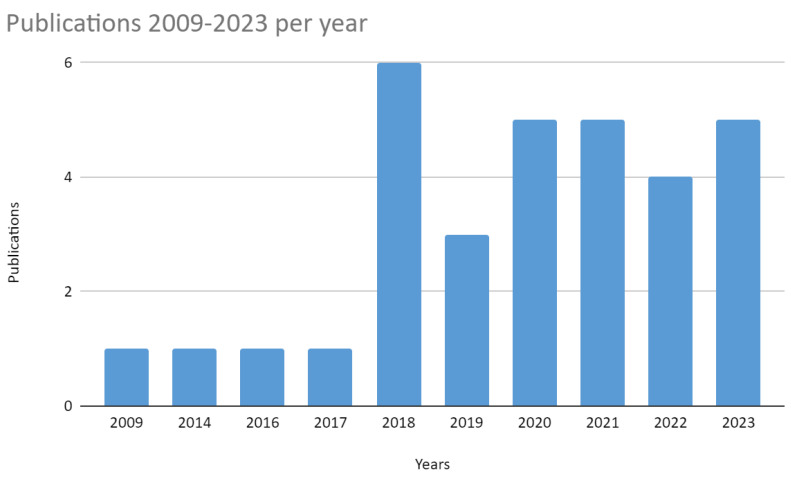
Publications by year.

**Table 1 ijerph-21-00413-t001:** Publications that related greenness, university campus, wellness or mental health.

Authors	Region/Year	Study Objective	Method	Sample	Results
Tian et al. [26]	Anhui and Shandong provinces, China, 2023	To investigate the correlations of greenness exposure with test anxiety among university students during COVID-19 lockdowns.	Cross-sectional study with perceived campus greenness (5-point Likert scale: quality, visibility, abundance, usage, accessibility) and Normalized Difference Vegetation Index (NDVI) for objective greenness	2609 university students	NDVI 1500-m correlated with lower test anxiety, physical activity may partially mediate this association. Increased campus greenness may alleviate test anxiety among Chinese university students.
Gulwadi et al. [4]	Turkey and the United States, 2019	To correlate objective and perceived greenness and restorative effects among students from two campuses in the USA and two campuses in USA.	Cross-sectional study for perceived greenness and quality of life survey (WHOQOL-BREF: psychological, physical, social, and environmental) and use of NDVI for objective greenness	1079 university students of business, design, and psychology	The space immediately next to buildings in which instruction takes place gain more significance when students spend longer hours in class, as that offers the closest access to greenspaces and enables a setting for social support.
Liu et al. [7]	University in Fuzhou, China, 2018	To develop a self-rated naturalness scale to measure perceived naturalness, and examine the association between self-rated naturalness and students’ restoration and health	Questionnaire with indicators for assessing perceived naturalness (self-rated naturalness scale SRNS developed at Fujian University) based on Chinese socio-cultural background in 8 university campus areas.	2550 university students of agriculture and forestry	Self-rated health and higher levels of restoration showed a positive relation with the perception of high natural attributes and the perception of natural form, both for men and women, even controlling for physical activity and social cohesion.
Liprini & Coetzee [21]	University of Pretoria, South Africa, 2017	To determine where students spend their free time on campus, how they perceive their on-campus green spaces and the extent they found these spaces restorative.	Questionnaire for Perceived restorativeness Scale (PRS a 26-item scale that measures constructs of Attention Restoration Theory, ART). Statistical analyses to determine which of the green spaces contributed most significantly to overall PRS scores.	286 university students	All green areas made significant contributions, but the campus’ botanical garden with a large amount of variety was the greatest contributor to Perceived Restoration Scale, followed by areas with presence of water.
Hipp et al. [8]	United States, 2016	To verify if green campus spaces provide restorative potential to university students and if students perceive it.	Students were surveyed with items on perceived greenness of campus, perceived restorativeness of campus, and the World Health Organization Quality-of-Life Scale (WHOQOL-BREF)	5683 students of three universities	Results indicate that those with higher perceived campus greenness report greater quality of life, a pathway significantly and partially mediated by perceived campus restorativeness.
Bernat et al. [15]	Lublin, Poland, 2022	To determine whether and to what extent the COVID-19 pandemic influenced the importance of recreation areas and the perception of landscape among students affected by lockdown.	Survey conducted in two stages, using online surveys. The first concerned with the perception of landscape and the second with the importance of recreational areas. (18 closed, single, and multiple-choice questions, a few with use of the 5-point Likert scale)	381 university students	Significance of recreation areas increased during the pandemic. The perception of landscape changed as well: the value of nature, scenic views, and therapeutic effect of the landscape began to be appreciated to a greater extent. Survey results indicate the need to ensure diversity of green areas and improve their accessibility, and to consider quiet areas, green mobility, and places of recreation.
Larson et al. [16]	United States, 2022	To assess how, and why, outdoor recreation and park use changed since COVID-19.	During the early stages of the pandemic, web survey measured students’ self-reported levels of emotional distress (open-ended responses coded and analyzed using conventional content analysis) and assessed potential demographic and contextual correlations to distress, including county-level per capita park area and greenness, using generalized linear models. Objective greenness measured by NDVI.	1280 students of four large public universities across the country	Reducing use of outdoor spaces due to lockdowns, concerns about viral transmission, and negative emotions. Those who maintained pre-pandemic park desired to be in nature, for improving mental and physical health. Emotional distress among students was widespread. Higher levels of emotional distress were associated with reducing park use and residing in counties with a smaller area of parks per capita.
Ha & Kin [19]	United States, 2021	To verify if more biodiverse green spaces have more restorative effects than conventional lawns in campus environments.	Four-group pre-post-test randomized experiment, and a 360-degree view video via an online platform survey. Students were randomly assigned to four experimental settings: (1) low-biodiversity with no sound, (2) high-biodiversity with no sound, (3) low-biodiversity with sound, and (4) high-biodiversity with sound. Restorative and mood status were evaluated using the restorative state scale (RSS) and the Short Form of The Profile of Mood States (POMS-SF).	319 university students	Neither the level of biodiversity nor natural sound had a significant effect on restorative and mood states. However, there was a statistically significant interaction effect between auditory and visual stimuli in mood states. The presence of natural sound with a high biodiversity environment showed lower negative mood states than the absence of natural sound. Results indicated that perceived plant species richness is positively associated with negative mood states. These discrepancies may be explained by poor biodiversity identification.
Oswald et al. [27]	Australia, 2021	To explore associations between the four states of mental health and protective factors relevant to young Australians and their mental health in the context of COVID-19.	Online survey to assess mental illness and mental well-being and cross-classify into four mental health states using 14-item self-reported mental health continuum-short form (MHC-SF) and the self-report Kessler Psychological Distress scale. Multinomial logistic regressions used to examine associations, adjusting for socio-demographic confounders.	1004 (participants 18–24 years old)	Protective factors for flourishing mental health were secure employment, use of screen to connect, high levels of hope and incidental and purposive contact with nature. Lack of green/blue space within walking distance was associated with languishing mental health, and lower neighborhood greenness was associated with all three sub-optimal mental health states. Young adults require dedicated mental health services to deal with current burden, also supported through preventive strategies which target mental health risk factors, like precarious employment, and enhance protective factors, such as urban green infrastructure.
Nazif-Munoz et al. [28]	El Paso, TX, USA, 2020	To assess the association between depression and residential greenness, brownness, and grayness.	Prospect cohort study. Depression measured with Patient Health Questionnaire-9 scale (PHQ-9) and greenness with NDVI. Data from the National Land Cover Database, for measures of land patterns: grayness and brownness. Structural equation models assessed the relationships of these land patterns to depression.	393 Nursing students (18–60 years old)	After adjusting for individual characteristics, at buffers 250 m, greenness was not associated with a decrease in the incidence rate ratios of depression; however, grayness and brownness were, respectively, associated with increases by 64% and decreases by 35%. It found a protective association between brownness [natural desert landscape] and depression.
Loder et al. [23]	Graz, Austria, 2020	To investigate the perceived greenness of college students’ home and study environments and its relation to mental health.	Online survey to assess perceived greenness at home and at university using questions on quality of and access to green space; mental health was measured with the WHO-5 well-being index. Uni- and multi-variate regression analyses used to analyze the data.	601 college students (24 years old)	The analyses revealed positive associations between perceived greenness at home and mental health, as well as perceived greenness at university and mental health. This adds more evidence to the existing literature that perceiving the environment as green is positively related to better mental health.
Loder et al. [29]	Graz, Austria, 2020	To verify the associations between objective or subjective neighborhood greenness and health.	Objective and perceived greenness assessed at home and at university. Health measures by questionnaire for mental health (WHO-5), IPAQ questionnaire for physical activity and sedentariness, and body mass index. Per location, quintile pairs of objective and perceived greenness were classified into underestimates, correct estimates, or overestimates.	377 university students	ANOVA models only showed poorer mental health for students underestimating greenness at university compared to those with correct estimates. Agreement between greenness scores at home but not at university was obtained, and mental health was related to the perception of greenness at university.
Srugo et al. [13]	Ontario, Canada, 2019	To investigate associations between school-based measures of greenness and students’ mental health in Canada.	Link participant responses from the 2016–2017 Ontario Student Drug Use and Health Survey to school-based features of the built environment. Measures of greenness with NDVI. Measures of mental health included: serious psychological distress [Kessler 6-item Psychological Distress Scale], self-rated mental health, suicide ideation, and suicide attempt.	6313 students (11–20 years)	Prevalence of serious psychological distress and low self-rated mental health was 16.7% and 20.3%, respectively. Suicide ideation reported by 13.5%, and 3.7% reported a suicide attempt. Quantity of greenness was similar between schools. In logistic regressions there was no association between objective school-based greenness and mental health, as assessed by multiple measures.
Dzhambov et al. [30]	Plovdiv, Bulgaria, 2019	To examine relations among restorative experience, mindfulness, rumination, and psychological resilience, linking residential greenspace to anxiety and depression symptoms.	Greenspace measured by NDVI and tree cover density. Symptoms of anxiety and depression measured with the Generalized Anxiety Disorder scale (GAD-7) and Patient Health Questionnaire (PHQ-9). Mediators assessed by self-report: perceived greenspace, restorative quality of the neighborhood, dispositional mindfulness, rumination, and psychological resilience. Structural equation modeling techniques used to test the theoretically indicated relations among variables.	529 undergraduate university students in health professions: medicine, pharmacy, and dentistry	Across different buffer sizes, higher greenspace was consistently associated with reduced scores on the anxiety and depression scales. This effect was partially mediated via several pathways. Specifically, higher NDVI 500-m was associated with higher perceived greenspace, and in turn, with higher restorative quality, and then with higher mindfulness, lower rumination, and greater resilience to stress, and consequently, with better mental health.
Dzhambov et al. [5]	Plovdiv, Bulgaria, 2018	To examine how different pathways between green/blue space and mental health can work together.	Residential greenspace measured by NDVI, tree cover density, percentage of green areas, and distance to green space. Blue space is measured by presence in the neighborhood. Mental health measured with General Health Questionnaire (GHQ-12). Mediators considered: perceived neighborhood green/blue space, restorative quality of the neighborhood, social cohesion, physical activity, noise and air pollution, and environmental annoyance. Structural equation modeling used to analyze data.	720 (18–35 years old) medical students	Higher NDVI within a 300 m buffer around the residence was associated with better mental health through higher perceived greenspace, leading to increased restorative quality, and subsequently to increased physical activity; through lower noise exposure, which in turn was associated with lower annoyance. Presence of blue space within a 300 m buffer did not have a straightforward association with mental health owing to competitive indirect paths.
Ibes et al. [9]	Williamsburg, USA, 2018	To investigate the psychological impact of two short green space interventions that integrate two proven approaches to stress reduction, mind–body skills, and nature exposure.	Two models of short, low-cost interventions were evaluated for their potential to mitigate chronic stress while providing interactions with nature, RESET (Release Everyday Stress and Enjoy Trails): first a 1 min “Digital detox nature stop”, second a 5 min. “Mindfulness in Nature trail”. coding schema and method for evaluation.	558 voluntary participants (undergraduate and graduate students, faculty, staff, and visitors)	Statistical and qualitative analysis of the participants reveals that these simple, low-cost interventions were instantly popular and well-received and had a positive psychological effect on 96% of participants who reported psychological impact, most commonly, relief from stress (82%).
Dzhambov et al. [31]	Plovdiv, Bulgaria, 2018	To investigate whether residential greenspace modified the effect of road traffic noise on general mental health in students.	Road traffic noise (Lden) level was calculated from the strategic noise map of the city. Objectivegreenspace tested: NDVI, tree cover density, and distance to the nearest green space. Mental health measured with the General Health Questionnaire (GHQ-12). Moderation analysis and the Johnson-Neyman (J-N) procedure were used to identify values along the continuous moderators.	399 (15–25 years students from two high schools and three universities)	Results indicated that living in a neighborhood deprived of trees enhanced the negative effect of noise, whereas in neighborhoods with higher tree cover density noise had no effect. Living in a less green neighborhood may enhance the negative effect of road traffic noise on mental health. This observed effect modification may not only be due to disrupted propagation of sound waves, but also to higher recreational quality in a greener environment.
Dzhambov [32]	Plovdiv, Bulgaria, 2018	Pilot study aimed to examine mechanisms/variables mediating associations between residential green/blue space and symptoms of anxiety/depression.	Cross-sectional and longitudinal. Students followed from the beginning to the end of the school year. Residential green space was defined by NDVI. Blue space assessed in the same buffers. Levels of anxiety/depression assessed using General Health Questionnaire (GHQ-12). mediator variables: residential noise and air pollution, environmental annoyance, perceived restorative quality of neighborhood, social cohesion, physical activity, and sleep disturbance.	109 students (18–35 years old of the Medical University)	Higher NDVI correlated with better mental health only indirectly through higher physical activity and restorative quality. Longitudinal data showed improved mental health but no significant effect of mediator variables. Similarly, blue space correlated with better mental health in all models, but physical activity and restorative quality were significant mediator variables only in the cross-sectional analysis.
Dzhambov et al. [33]	Plovdiv, Bulgaria, 2018	To compare single and parallel mediation models, which estimate the independent contributions of different paths, in the pathway from greenspace to mental health.	Objective greenspace defined by NDVI, Soil Adjusted Vegetation Index, tree cover density and distance to nearest greenspace. Self-reported measures of availability, access, quality, and usage of greenspace were used. Mental health measured with General Health Questionnaire (GHQ-12). Potential mediators considered in single and parallel mediation models: restorative quality of the neighborhood, neighborhood social cohesion, commuting and leisure time, physical activity, road traffic noise annoyance, and perceived air pollution. Four models tested with serial mediation components: [1] restorative quality → social cohesion; [2] restorative quality → physical activity; [3] perceived traffic pollution → restorative quality; [4] and noise annoyance → physical activity.	399 (15–25 years of age)	There was no direct association between objectively measured greenspace and mental health. For the 500-m buffer, the tests of the single mediator models suggested that restorative quality mediated the relationship between NDVI and mental health. Tests of the serial mediation models showed that higher restorative quality was associated with more physical activity and more social cohesion, and in turn with better mental health. As for self-reported greenspace measures, single mediation through restorative quality was significant only for time in greenspace; however, serial mediation through restorative quality and social cohesion/physical activity was indicated for all self-reported measures except for greenspace quality.
Asim et al. [12]	Himachal Pradesh, Himalayas, India, 2023	To explore the role of the built environment of an academic campus in influencing students’ perception of mental restoration and well-being.	Survey to identify activity hotspots on campus, and a neuropsychological study involving mobile EEG and Perceived Restorativeness Scale (PRS) at these hotspots to investigate the role of built environment aspects on restorative approach/neglect behavior and perception of built environment. The study relies on prominent theoretical constructs in the area, namely Attention Restoration Theory (ART), Stress Reduction Theory (SRT).	429 college students (sub-sample 22 students)	Environmental aspects, such as vegetation, novelty/uncertainty, acoustic ambience, and expansive vistas, are positively associated with favorable approach behaviors and normalized average alpha brain waves, while other elements, such as presence of high density of buildings may be related to avoidance behaviors. Greenery and Sky had the most positive relationship with the normalized alpha, presence of buildings revealed negative relationships. Natural components such as grass, sky, and filtered and diffused light boost people’s perceptions of an environment’s restorative characteristics.
Bardhan et al. [11]	Public university, SE United States, 2023	To conduct one of the first longer-term investigations of daily nature exposure and mood with a mobile app as part of the NatureDose™ Student Study (NDSS).	The NatureDose™ app uses GPS and a phone’s sensors to evaluate levels of nature surrounding the person with the NatureScore™ measure, based on more than 30 remotely sensed datasets, geo-located health records, and machine learning models. College students’ nature exposure monitored for eight weeks, their mood states calculated weekly using the Positive and Negative Affect Schedule (PANAS). Observational associations between average nature exposure and mood levels across the study period used mixed-effect linear regression models adjusting for gender and allergies.	97 university students (18–22 years old)	Positive association between nature exposure and positive mood. Findings supported by sensitivity analyses with ANOVA models on average NatureDose™ levels. Average nature exposure was 71 min per day. Study supports the utility of using NatureDose™ as an app-based tracking tool for objective nature exposure measurement and reinforces previous findings on the associations between nature exposure and positive mood states. Greening university campuses could support college students’ mental health.
Lemyre et al. [34]	Oxford and Southampton, United Kingdom, 2023	To investigate use of neighborhood greenspace access and well-being.	During the tail end of the third ‘lockdown’, an online questionnaire was used to better understand mental well-being in relation to use of outdoor green space (assessed using a 5-item Likert scale). Analyses include descriptive results of indicators and hierarchical multiple linear regression models.	424 university students	Quality of greenspace had a greater importance on mental well-being than use and quantity of greenspace, even when controlling for sociodemographic factors. Also, neighborhood greenspace quality contributed to well-being above and beyond sociodemographic, physical activity and social support. This result held true even among students with prior mental health difficulties.
Ning et al. [35]	Changchun, Jilin Province, China, 2023	To explore the transient recovery effects of four types of campus environments, square space, dense forest, sparse forest, and waterfront.	Field experiment students, measuring systolic blood pressure, diastolic blood pressure, and heart rate as physical parameters to assess stress recovery. Respondents also reported scores (Restorative components scale -RCS-based on environmental restorative scale–PRS) about their personal feelings in questionnaires to evaluate their psychological states.	60 university students of Landscape Architecture	Physiological and psychological indicators responded to the brief natural exposure [5 min], and physiological and psychological health was restored. Only the recovery amounts of psychological indicators were significantly different in waterfront space, dense forest space, and sparse forest space. Brief exposure in the waterfront space was the most beneficial to students’ psychological health recovery.
Boyd [36]	University of Sheffield, United Kingdom, 2022	To examine the participants’ interactions with nature and experience of the university campus and design interventions.	Mixed methods approach, with statistical analysis and focus group discussion. Use of a green prescription style activity and a specially designed mobile phone app (Shmapped) to evaluate how participants experienced the 30 days interventions.	55 students completed the experiment	Findings qualify research into young adult’s experience of urban green spaces and their tangible connection to plants such as trees. Policy and practice implications include the requirement for a coherent approach to understanding the place-attachment aspects to nature in the university environment. There is a need for collaborative well-being interventions and urban green space development within the UK context.
Stepansky et al. [37]	Private university, NE United States, 2022	To verify if engagement in the use of green spaces improved health and well-being.	Pilot study with lived experience of two students and investigation of perceptions of a university campus therapeutic sensory garden on the quality of life (WHOQOL) based on the utilization of the green space. Quantitative and qualitative measures assessed student use of therapeutic sensory garden and perceived quality of life related to social, mental, and physical well-being.	Two university students	Findings from quantitative and qualitative analysis of the lived experiences of two students are consistent with that of current evidence literature indicating a positive association between time spent in a green space/natural environment and perceptions of quality of life. Pilot study findings present structure and hypotheses related to time utilization, anticipated outcomes, and active ingredients for therapeutic sensory garden intervention with larger sampled groups of university students.
Asim et al. [12]	Haridwar District, Himalayan, India, 2021	To understand the role of containment zone built environments in the prevalence of anxiety and depression.	Questionnaires to participants in three hostels in the containment zone. Linear regression between Built Environment Variable Score with Center for Epidemiologic Studies-Depression (CES-D) and Generalized Anxiety Disorder Scale (GAD-7).	432 University students	Results revealed that students living in rooms with access to quality window views overlooking greenery and sky in addition to the presence of indoor plants and portrait/artworks, are at lower risk of depression and severity of anxiety.
Kim et al. [14]	Chungbuk National University, campus forest, South Korea, 2021	To examine the psychological effects of forest activities in a campus forest.	Pre and post-test to evaluate psychological effect of forest activities in a campus. The Profile of Mood State (POMS) questionnaire, the Concise Measure of Subjective Well-Being (COMOSWB), and the Stress Response Inventory (SRI-MF) were administered to assess psychological effects.	38 university students [mean age 22, with no diagnosis of severe stress or depression and no drug or alcohol abuse]	This study revealed that participants in the forest activities intervention group had significantly positive increases in their mood, stress response, and subjective well-being, compared with those of control group participants who did not partake in any forest activities.
Souter -Brow et al. [24]	Auckland, New Zealand, 2021	To explore the potential of salutogenic design as a stress-reducing health promotion tool for ‘apparently well’ people in a workplace setting.	Randomized controlled trial compared ‘apparently well’ participants into intervention groups, sensory garden (SG), urban plaza (UP), and a control group (CG). SG and UP participants had ‘appointments outdoors’ once weekly for 4 weeks. All tested for salivary cortisol, perceived well-being, productivity, perceived stress, nature relatedness pre- and post-intervention; data analyzed using generalized linear models.	164 (18–65 years old staff and students)	Significant intervention effects were observed for salivary cortisol, well-being and productivity. Although not significant, a surprising trend towards negative effects of the urban plaza on well-being, productivity, and perceived stress were observed when compared to the sensory garden group. This study suggests a sensory garden effectively reduces stress, enhances well-being and improves productivity of ‘apparently well’ people in the workplace.
Malekinezhad et al. [22]	Malaysia, 2020	To investigate how the perception of characteristics of natural space, directly and indirectly, affects restoration experience within the area of green outdoor landscape.	Test of associations through application of partial least squares with structural equation modeling (PLS-SEM), inputting data from a sample of university students. Perceived Sensory Dimension (PSD) and Restorative Outcome Scale (POS-6 items) were used in measuring restoration experience.	550 students (five universities)	The effect of perceived sensory dimensions (PSD) on perceived restorativeness leading to better explanation of restoration experience. Perceived landscape characteristics of PSD enhance restoration experience and is a mediator of the relationship between perceived campus greenness and students ‘quality of life.
Reese et al. [38]	Pacific Northwest, United States, 2020	To explore the role that campus greenspaces might play in preventing stress and promoting student health on college campuses	Qualitative study using photovoice (community-based participatory research) in crafting a vision for how natural spaces might be maintained and developed to promote health. Five themes emerged: mental and physical health, community spaces, sustainable infrastructure design, preserving natural habitat and history, homelessness.	72 undergraduate students	Aspects of the student vision may be incorporated into future campus environmental planning efforts. Additional research is needed in determining whether community-based greenspace planning on college campuses can positively impact the extent to which students access natural spaces for the purposes of alleviating stress and promoting health.
Krasny & Delia [39]	The Cornell campus at Cornell Natural area, USA, 2014	To understand how university students experience engagement in campus natural area stewardship and related policy discussions.	Adaptative co-management (ACM) “learning from doing” concept guided encompassing social learning, social capital, and shared action. Student semi-structured interview and focus group discussion in “an experiment effect”.	Ten undergraduate students	Authors found that three conditions explain student engagement in the adaptive co-management process: the presence of a pre-existing student organization that had built bonding social capital and was committed to campus natural area stewardship, openness to multiple stakeholder viewpoints and commitment to action on the part of the university administration, and the presence of a crisis that spurred emotions and action. Student organizations can contribute to an adaptive co management process, consistent with university and campus sustainability.
Lau & Yang [40]	China. Hong Kong University, 2009	To study the potential role of the introduction of healing gardens to a compact campus in creating a health-supportive and sustainable campus environment.	Case study of introducing healing gardens and observation of design and use patterns. Questionnaire survey on green space perception and usage, natural view, and preferred resting points.	33 undergraduate and postgraduate students	Therapeutic benefits of a healing garden are recognized by staff and students. Because of the small campus size, green spaces are reduced and do not encourage large groups of people but provide nature like view from surrounding paths and windows, providing physiological benefits such as stress attenuation and attention restoration. Natural settings such as courtyard, atrium and green roof gardens, and green walls are suitable for a small campus.
Total				27,955	

Source: Organized by the authors.

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
