# Peer review of "Natural Environments in University Campuses and Students’ Well-Being"

_ijerph, 2024, doi:10.3390/ijerph21040413_

Round 1

Reviewer 1 Report

Comments and Suggestions for Authors

PROPOSALS FOR IMPROVEMENT

The article is of interest and topicality since it addresses the relationship between the emotional well-being or mental health of university students and the natural environments existing on these university campuses.

However, the following considerations should be taken into account in the structure of the article in order to improve it.

Line 136: Section 2 Materials and methods should be structured as follows:

Method

Although line 165 and Figure 1 indicate that PRISMA guidelines were used, this should be specified in this first section.

Search procedure

The date range in which the searches were carried out should be indicated and the text should be structured as follows: the databases used for this purpose, as well as the descriptors and Boolean operators used. Also the inclusion and exclusion criteria.

These questions mentioned above should be included in a table with the search strategies, specifying both the descriptors used and the places where they were searched (Title, keywords, Abstract...), in order to make it more exemplified.

Data extraction instrument

Indicate the analysis technique used (documentary analysis, etc.), the instrument used, for example, information matrix, and the data on which relevant information was extracted (author, year of publication, origin, etc.).

3. Results

Identification of the articles

In this section, the process followed is presented in Figure 1: search and results according to PRISMA guidelines.

Characteristics of the studies

Brief narration of those presented in Table 1, together with the table itself.

In Table 1, in the "Method" column, it is convenient to cite (instrument name, author and year of validation) of the instruments used in the study, especially when they are standardized questionnaires.

Discussion and Conclusions: nothing to object to.

References

They should be in APA 7th edition format.

Author Response

Line 136: Section 2 Materials and methods should be structured as follows:

Method

Although line 165 and Figure 1 indicate that PRISMA guidelines were used, this should be specified in this first section.

Response: It was included on lines 156 and 157.

Search procedure

The date range in which the searches were carried out should be indicated and the text should be structured as follows: the databases used for this purpose, as well as the descriptors and Boolean operators used. Also the inclusion and exclusion criteria.

Response: All this information was included in lines 167 to 182.

These questions mentioned above should be included in a table with the search strategies, specifying both the descriptors used and the places where they were searched (Title, keywords, Abstract...), in order to make it more exemplified.

Response: We improved the description to make it more understandable in line 185 to 189..

Data extraction instrument

Response: Extraction was made mannually and described on line 188 and 189.

Indicate the analysis technique used (documentary analysis, etc.), the instrument used, for example, information matrix, and the data on which relevant information was extracted (author, year of publication, origin, etc.).

Response: It was described on lines 188 and 189.

3. Results

Identification of the articles

In this section, the process followed is presented in Figure 1: search and results according to PRISMA guidelines.

Characteristics of the studies

Brief narration of those presented in Table 1, together with the table itself.

In Table 1, in the "Method" column, it is convenient to cite (instrument name, author and year of validation) of the instruments used in the study, especially when they are standardized questionnaires.

Response: The intrument names were described on the table as they were mentioned in the articles analyzed.

Discussion and Conclusions: nothing to object to.

References

They should be in APA 7th edition format

Response; The references follow the instructions in the periodical guidelines for manuscript preparation.

Reviewer 2 Report

Comments and Suggestions for Authors

The authors conducted the review on the literature on campus greenery and student well-being. The review of 32 articles indicates the positive relationship between these two constructs. The manuscript is light on analysis and theoretical underpinnings. I am suggesting a few action items below.

1.        The authors need to identify theoretical underpinning supporting the relationship between the two constructs and create an overarching framework conceptualizing factors affecting student well-being.

2.        The authors should comb through the identified articles, and perform a thorough analysis by identifying what has been studied and what are areas that need academic attention.

3.        In terms of the method for the literature review, the authors need to justify the selection of the time period (2009- 2-23) and the research libraries.

Comments on the Quality of English Language

The quality of English is good.

Author Response

  1. The authors need to identify theoretical underpinning supporting the relationship between the two constructs and create an overarching framework conceptualizing factors affecting student well-being.

Response: The theoretical underpinning supporting the relationship between the two constructs was improved and is described on lines 121 to 144 of the new version.

  1. The authors should comb through the identified articles, and perform a thorough analysis by identifying what has been studied and what are areas that need academic attention.

Response: A comb in the results was undertaken and included in lines 282 to 307.

  1. In terms of the method for the literature review, the authors need to justify the selection of the time period (2009- 2-23) and the research libraries.

Response: The selection had no initial cut-off date until february 2023 when we did the search. This is described on lines 167 and 168. We did not use an initial cut-off date because we wanted to verify when studies on the theme started to appear in scientific literature.

Research libraries are justified in lines 158 and 167 to 170.

Reviewer 3 Report

Comments and Suggestions for Authors

Natural Environments in University Campuses and students’ well-being

Review for IJERPH

l.62: The authors mention the fact that university students face a range of academic, interpersonal, financial and cultural challenges, and that these challenges can lead to psychological and other health problems. It is necessary for authors to document these elements. What studies have proven this? On the basis of which article do the authors base their statement?

l.65: The authors state as truth the fact that many of these young adults attending university are facing their first extended period away from home. What figures are there to support this statement? In France, for example, the percentage of Toulouse students enrolled in one of Toulouse's universities is significantly higher than that of students from elsewhere.

l.72: The authors go on to argue that digital overload exacerbates chronic stress and anxiety, as students tend to spend many hours a day on smartphones, tablets, computers and cell phones. What does this have to do with the paragraph just before, and on the basis of what study do the authors base this statement?

l.74: Same remark when the authors mention that when students don't live on campus, they tend to spend most of the day watching TV. What reference?

l.75: Why do the authors state that studies suggest that the prevalence and severity of mental health problems in people with mental disabilities is higher than in people with physical disabilities. Is this a variable that will be incorporated into the literature review for this article? If so, it should be developed and argued. If not, no interest.

l.107: The authors state that their first objective was to focus the study on Latin American campuses. Why did they do this? What is the theoretical basis for concentrating this interest in Latin America rather than elsewhere?

The problem statement is unclear. What is the research question, what is the theoretical controversy in which this study is embedded? What is the general hypothesis and what are the operational hypotheses?

The literature review was based on keywords related to well-being, nature and the university. As a result, the comments made about the other variables mentioned above are repeated. Why mention disability, for example? Or the fact of being more or less connected?

The large analytical table should be appended.

The analysis of the articles is not well organized. What are the authors' hypotheses? What do they hope to extract in terms of innovation?

Author Response

l.62: The authors mention the fact that university students face a range of academic, interpersonal, financial and cultural challenges, and that these challenges can lead to psychological and other health problems. It is necessary for authors to document these elements. What studies have proven this? On the basis of which article do the authors base their statement?

Response: We have included references to these statements on lines 76 to 82.

l.65: The authors state as truth the fact that many of these young adults attending university are facing their first extended period away from home. What figures are there to support this statement? In France, for example, the percentage of Toulouse students enrolled in one of Toulouse's universities is significantly higher than that of students from elsewhere.

Response: We included the reference on line 65.

l.72: The authors go on to argue that digital overload exacerbates chronic stress and anxiety, as students tend to spend many hours a day on smartphones, tablets, computers and cell phones. What does this have to do with the paragraph just before, and on the basis of what study do the authors base this statement?

Response: References and further text was included in lines 72 to 76.

l.74: Same remark when the authors mention that when students don't live on campus, they tend to spend most of the day watching TV. What reference?

Response: This remark is not in our text.

l.75: Why do the authors state that studies suggest that the prevalence and severity of mental health problems in people with mental disabilities is higher than in people with physical disabilities. Is this a variable that will be incorporated into the literature review for this article? If so, it should be developed and argued. If not, no interest.

Response: This variable is not in our text.

l.107: The authors state that their first objective was to focus the study on Latin American campuses. Why did they do this? What is the theoretical basis for concentrating this interest in Latin America rather than elsewhere?

Response: The explanation for our interest in Latin America is given on lines 111 to 114.

The problem statement is unclear. What is the research question, what is the theoretical controversy in which this study is embedded? What is the general hypothesis and what are the operational hypotheses?

Response: Thank you for remark. We have improved the theoretical basis on lines 121 and 144 and our hypothesis on lines 145 and 146.

The literature review was based on keywords related to well-being, nature and the university. As a result, the comments made about the other variables mentioned above are repeated. Why mention disability, for example? Or the fact of being more or less connected?

Response: Disability was not mentioned in our study. The fact of being connected in excess was associated to anxiety and depression in some of the articles analyzed.

The large analytical table should be appended.

Response: The table represents the results of the systematized review and for this reason it was not appended.

The analysis of the articles is not well organized. What are the authors' hypotheses?

Response: The hypothesis is on lines 145 and 146.

What do they hope to extract in terms of innovation?

Response: The innovation expected was included on lines 149 to 154.

Round 2

Reviewer 2 Report

Comments and Suggestions for Authors

I am glad to see the improvement made. However, I'd like the authors to further improve their discussions on the findings by focusing on how their literature review Attention Restoration Theory (ART) and Stress Restoration Theory (SRT) in the context of natural environments in university campuses. It seems to me that the reviewed articles shed light on the different conceptualizations of green spaces (e.g., size, location, composition), functions of green spaces, and governance of the spaces. The authors could build on the current literature and create an overarching framework capturing the impact of green spaces on students' well-being.  In addition, the authors should also highlight the areas that need future academic attention to deepen our understanding of the impact of green spaces on students' well-being.

Author Response

We thank you for the suggestions. We have included further information on the Theories regarding students' restoration and the gaps in the literature that need further attention in studies. 

The additional text in in red in the revised version.

Reviewer 3 Report

Comments and Suggestions for Authors

Thank you

Author Response

We also thank you for the important suggestions made.